# A Case of Mediastinal Tuberculous Lymphadenitis in a Chronic Dialysis Patient Diagnosed by Endobronchial Ultrasound-Guided Transbronchial Needle Aspiration (EBUS-TBNA)

**DOI:** 10.3390/medicina59040677

**Published:** 2023-03-29

**Authors:** Hiromi Nagashima, Kazuyuki Abe, Yukihiro Owada, Kazuhiro Yakuwa, Hiroshi Katagiri, Shinji Chiba, Ami Matsumoto, Masachika Akiyama, Yu Utsumi, Makoto Maemondo

**Affiliations:** 1Department of Respiratory Medicine, Iwate Medical University, Iwate 028-3694, Japan; 2Department of Respiratory Medicine, Iwate Prefectural Yamada Hospital, Iwate 028-1352, Japan

**Keywords:** mediastinal tuberculous lymphadenitis, fever of unknown origin, hemodialysis, endobronchial ultrasound-guided transbronchial needle aspiration (EBUS-TBNA)

## Abstract

A 54-year-old woman on dialysis due to chronic renal failure had a fever lasting 2 weeks and was referred to a hospital. Non-enhanced CT and blood tests showed no remarkable findings. She was hospitalized and received an antibacterial drug. Although she was discharged after the fever subsided, she was hospitalized again due to a fever a few days later. A contrast-enhanced CT revealed mediastinal lymphadenopathy, and she was transferred to our hospital for a bronchoscopy. Endobronchial Ultrasound-Guided Transbronchial Needle Aspiration (EBUS-TBNA) for subcarinal lymph nodes was performed in our hospital. The Polymerase Chain Reaction (PCR) test of the obtained specimen was positive for mycobacterium tuberculosis, and histologically, caseous granulomas were found in the specimen. She was diagnosed with mediastinal tuberculous lymphadenitis, and HREZ (isoniazid, rifampicin, ethambutol, and pyrazinamide) treatment was started. The fever subsided immediately, and she was discharged from our hospital 2 weeks after the initiation of treatment. Thereafter, she received treatment as an outpatient. Since the use of a contrast medium was complicated by dialysis, a non-enhanced CT was performed at first, and it was difficult to make a diagnosis from this. We report this as an informative case that could be diagnosed with EBUS-TBNA, which was easily performed on a patient weakened by prolonged fever and dialysis.

## 1. Introduction

Although differential diagnosis of tuberculosis for patients with a fever of unknown origin is important, extrapulmonary tuberculosis is not easy to diagnose. In the case of tuberculous lymphadenitis, the most common type of extrapulmonary tuberculosis, lesions are often observed in the cervical lymph nodes. About 0.18% of all tuberculosis cases, commonly found in immune-deficient children, have the primary lesion in the mediastinal lymph nodes [1].

In this case, we diagnosed mediastinal tuberculous lymphadenitis in a dialysis patient by performing endobronchial ultrasound-guided transbronchial needle aspiration (EBUS-TBNA). In this case, the patient was undergoing dialysis, and a contrast-enhanced CT scan was not readily available. Mediastinal tuberculous lymphadenitis in adults is rare. This case suggests that the diagnosis was delayed due to a delay in contrast-enhanced CT.

Furthermore, EBUS-TBNA was an effective tool to aid in diagnosis, so we report this case.

## 2. Case

The patient was a 54-year-old Japanese woman. She smoked 20 cigarettes a day from age 20 to 54. She had had chronic renal failure since age 47 and had been on dialysis twice a week since age 51. The cause of chronic renal failure was unknown. She has a history of hypertension but has not taken steroids or immunosuppressants. There was no family history of tuberculosis or cancer. The patient had a fever lasting 2 weeks before visiting the previous hospital. However, she had no shortness of breath or chest pain. No cervical or axillary lymph nodes were palpable. Her cough and sputum were mild. Blood tests showed elevated C-reactive protein (CRP) and White Blood Cell (WBC). In the hospital, a chest X-ray and CT scan showed no abnormal shadow in the lung field. Additionally, pleural effusion could not be confirmed (Figure 1 and Figure 2A). She was hospitalized at that hospital and was started on ceftriaxone, 2 g/day, empirically. The culture tests of sputum, blood, and urine were negative for bacteria. Thereafter, the fever subsided, and she was discharged 10 days after admission. She developed a fever again a few days after discharge. The fever continued. She was admitted to the previous hospital again for further investigation and treatment of a fever of unknown origin. Even during the second hospitalization, sputum, blood, and urine culture tests were negative for bacteria. The fever did not subside even after the administration of cefotiam hydrochloride at a dose of 500 mg/day. The interferon gamma release assay test (T-SPOT.TB) showed a positive result (A antigen and B antigen were more than 50), although smears and cultures for the acid-fast bacillus test and tuberculosis PCR tests for sputum, spinal fluid, urine, blood, and gastric fluid were all negative. To diagnose her fever of unknown origin, a contrast-enhanced CT was performed after adjusting her dialysis. Contrast-enhanced CT scans taken 39 days after the second admission showed enlarged lymph nodes below the tracheal bifurcation. Lymph nodes were partially calcified. (Figure 2B). Retrospectively, we could detect swollen mediastinal lymph nodes in the previous CT (Figure 2A). The CT findings showed no other abnormal findings that could cause fever. She was transferred to our hospital 60 days after her second hospitalization in the previous hospital for the purpose of a biopsy in the mediastinal lymph nodes.

At the time of transfer, the body temperature was 38.9 °C. The blood pressure was 142/99 mmHg and the heart rate was 108/min. Physical examination showed no rales or abnormal cardiac sounds. Again, there were no palpable lymph nodes on the body surface. Laboratory data showed WBC 23,100/μL, BUN 63.4 mg/dL, Cre 9.72 mg/dL, CYFRA 7.6 ng/mL, PRO-GRP 345 pg/mL, and sIL-2R 3140 U/L (Table 1). PRO-GRP was determined to be rising due to renal failure.

After transfer, the patient’s status had deteriorated due to a fever that had persisted for 2 months. She could not eat enough. As a result, her weight was reduced by 3 kg in two months. She was lying on the bed all day.

Since a high level of soluble IL-2 receptor was observed in the blood test, a differential diagnosis between malignant lymphoma and tuberculous lymphadenitis was needed. Sarcoidosis is also a differential diagnosis, but since angiotensin-converting enzyme (ACE) was within the normal range and there were no ocular symptoms, there was no strong suspicion. Lung cancer was not strongly suspected because there were no findings in the lung field on CT.

A conference was held with a respiratory surgeon. Because the patient’s condition was deteriorating, the mediastinoscopic lymph node biopsy was abandoned. EBUS-TBNA was performed 2 days after the transfer. Bronchoscopy revealed no abnormal findings without the dull formation of the carina (Figure 3). A needle aspiration biopsy of the subcarinal lymph nodes was performed using EBUS-TBNA. The disease was diagnosed as mediastinal tuberculous lymphadenitis because the PCR test and culture of the obtained specimen were positive for Mycobacterium tuberculosis. Histopathological findings of the lymph nodes showed epithelioid cell granulomas and multinucleated giant cells compatible with tuberculosis (Figure 4). Treatment with daily isoniazid 300 mg/day and daily rifampicin 450 mg/day, plus ethambutol 750 mg/day and pyrazinamide 1.2 g/day on twice-weekly dialysis days, was started immediately. Mycobacterium tuberculosis detected in this patient had good drug sensitivity to anti-tuberculosis drugs. The fever subsided immediately. However, itching appeared during the treatment, which was thought to have been caused by the drugs. We temporarily reduced anti-tuberculosis drugs and treated itching with anti-allergic drugs. After that, the itching improved, and the treatment with anti-tuberculosis drugs was continued. The patient’s weight gradually increased, and her status improved. After that, she was discharged from the hospital. Three months after the initiation of treatment, a CT showed a reduction in the size of the mediastinal lymph nodes (Figure 2C). The treatment was continued for 9 months after initiation, and there was no recurrence after the treatment.

## 3. Discussion

This case of a dialysis patient with a fever of unknown origin was finally diagnosed as mediastinal tuberculous lymphadenitis by contrast-enhanced CT and EBUS-TBNA. A classical fever of unknown origin is defined as a fever of 38.3 °C or more that lasts for more than 3 weeks, the cause of which cannot be diagnosed despite 3 days of inpatient investigation or three outpatient consultations [2]. The percentage of cases of fever of unknown origin due to infectious diseases or malignant tumors has declined with advances in clinical tests and image diagnosis. Infectious diseases accounted for 36% of cases of fever of unknown origin in the 1950s, dropping to 24.5% in the 1990s [3]. In a 2013 report, the percentage was little changed at 23.1% [4]. Generally, a contrast-enhanced CT scan is a useful and informative test to determine the cause of a fever of unknown origin and should be employed as early as possible. However, it is difficult to perform in patients with severe renal dysfunction or those undergoing dialysis because dialysis is required immediately after the test. Miliary tuberculosis is an infectious disease that has long been linked to and is a common cause of fevers of unknown origin. Extrapulmonary tuberculosis other than miliary tuberculosis has also been considered to be a cause of fever of unknown origin [5]. Since the 1970s, extrapulmonary tuberculosis has been reported to occur more frequently in immunocompromised individuals [6]. In the United States, the acquired immunodeficiency syndrome epidemic coincided with a resurgence of tuberculosis incidence and high rates of disseminated and extrapulmonary disease [7]. A report has investigated the relationship between clinical symptoms of tuberculosis and CD4 cell counts in human immunodeficiency virus (HIV)-infected patients. In this study, decreased CD4 cells were associated with increased rates of extrapulmonary tuberculosis [8]. It has been reported that dialysis patients have a high proportion of cases of extrapulmonary tuberculosis, many cases of tuberculous lymph nodes, and many cases with a poor prognosis [9,10]. Cell-mediated immunity is reduced in dialysis patients. Decreased cell-mediated immunity allows Mycobacterium tuberculosis to grow in organs such as lymph nodes that are unsuitable for growth. Therefore, it is thought that extrapulmonary tuberculosis will increase [9].

Early diagnosis and early initiation of treatment are necessary to improve the prognosis, but diagnosis is often difficult. Dialysis patients from Morocco also reported that it takes a long time from the appearance of symptoms of tuberculosis to the diagnosis and initiation of treatment [11]. In the case of pulmonary tuberculosis, coughing, expectoration of sputum, and the appearance of new shadows in the lung fields are the triggers for suspicion of tuberculosis. In the case of tuberculous pleurisy accompanied by pleural effusion, pleural effusion that does not decrease even when the DW is lowered is a trigger for suspicion of tuberculous pleurisy. In both cases, a chest X-ray is diagnostic. However, chest X-rays often cannot identify abnormalities when the lesion is in the tracheal bifurcation, as in this patient. Even when simple imaging is used in a CT scan, early-stage identification of enlarged lymph nodes at the tracheal bifurcation is difficult. In our report, no abnormality was detected on the chest X-ray or non-enhanced CT. It is difficult to determine mediastinal lymphadenopathies with non-enhanced CT scans compared to contrast-enhanced CT scans. The diagnosis of mediastinal tuberculosis was inevitably delayed because the patient was on dialysis and could not easily undergo contrast-enhanced CT scans.

Previously, mediastinoscopy under general anesthesia was required to perform a biopsy of the mediastinal lymph nodes; however, the test was physically demanding for dialysis patients who are in poor general health. On the other hand, EBUS-TBNA can be performed under local anesthesia and is comparatively less physically demanding. In one study, EBUS-TBNA was performed on 156 patients with mediastinal lymph node tuberculosis. The culture was positive in 46% of the cases, pathological results consistent with the histology of tuberculosis were found in 86% of the cases, and 94% of the cases were diagnosed [12]. In our current report, the patient could undergo EBUS-TBNA without complications, and treatment was started 2 days after the procedure. If a dialysis patient is suffering from a fever of unknown origin, a differential diagnosis of mediastinal tuberculous lymphadenitis should be performed proactively, as the frequency of lymph node tuberculosis is higher in dialysis patients compared to the general population. Contrast-enhanced CT scans and EBUS-TBNA are essential diagnostic tools. However, EBUS-TBNA is less operable than a bronchoscope, and it is difficult to stop bleeding, so care must be taken. In addition, there is a report of a complication that occurred after puncture with EBUS-TBNA for mediastinal tuberculous lymphadenitis. In that case, a transient smear-positive, bloody sputum appeared after the examination [13]. Although it is complicated to perform contrast-enhanced CT on a dialysis patient, contrast-enhanced CT should be actively performed when examining a patient with a fever of unknown origin in order not to miss the diagnosis of extrapulmonary tuberculosis. In the future, it will be important to prevent the onset of tuberculosis in dialysis patients. A report from Canada used the interferon-gamma release assay test to compare a group screened for tuberculosis infection with an unscreened group. Of the 1790 screened patients, 152 were treated for latent tuberculosis infection (LTBI). As a result, it is reported that the incidence of active tuberculosis has decreased [14]. This case was not screened for tuberculosis infection at the time of initiation of dialysis. If this patient had been screened for tuberculosis infection, she could have been treated for LTBI. Had she been treated for LTBI, she might not have developed tuberculosis. Dialysis patients spend a long time in the dialysis room. There are many other dialysis patients in the dialysis room. If there is a tuberculosis patient in the dialysis room, many dialysis patients may become infected. The prevention of tuberculosis development in dialysis patients is important. However, LTBI treatment requires attention to side effects. There was a report of LTBI treatment for patients on dialysis. In this report, 365 patients were treated for LTBI. A total of 298 (81.6%) patients reported successful treatment. While many patients were successfully treated, 21.1% of patients experienced grade 3–4 adverse events or any grade of rash, and side effects should be monitored [15]. Although this patient was cured, extrapulmonary tuberculosis in dialysis patients often has a poor prognosis, so caution is required. This case has many lessons.

## 4. Conclusions

Dialysis patients have a high proportion of cases of extrapulmonary tuberculosis, many cases of tuberculous lymph nodes, and many cases with a poor prognosis. Although it is complicated to perform contrast-enhanced CT on a dialysis patient, contrast-enhanced CT should be actively performed when examining a patient with a fever of unknown origin in order not to miss the diagnosis of extrapulmonary tuberculosis. EBUS-TBNA was useful for the diagnosis of mediastinal tuberculous lymphadenitis in patients who were weakened by long-term fever and dialysis.

## Figures and Tables

**Figure 1 medicina-59-00677-f001:**
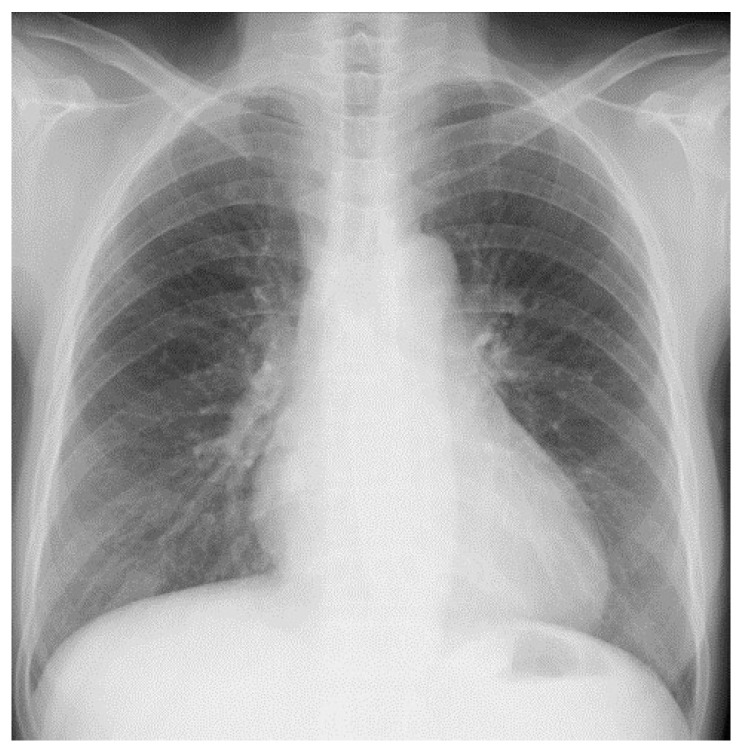
Chest X-ray image on admission to the previous hospital. No abnormal findings were observed.

**Figure 2 medicina-59-00677-f002:**
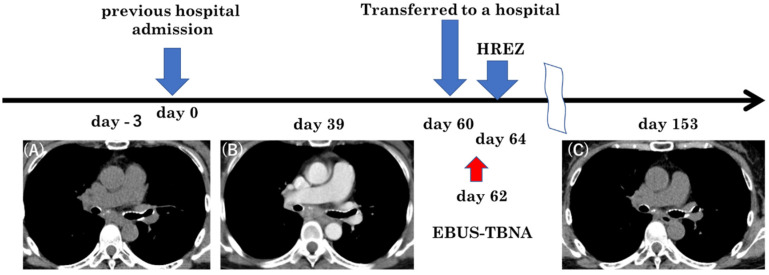
Chest CT scan revealing enlargement of subcarinal lymph nodes, initially undetected. (**A**) An increase in the size of the lymph nodes when treated as a fever of unknown origin. (**B**) Shrinkage of lymph nodes after anti-tuberculous treatment. (**C**) HREZ: isoniazid, rifampicin, ethambutol, pyrazinamide. EBUS-TBNA: endobronchial ultrasound-guided transbronchial needle aspiration.

**Figure 3 medicina-59-00677-f003:**
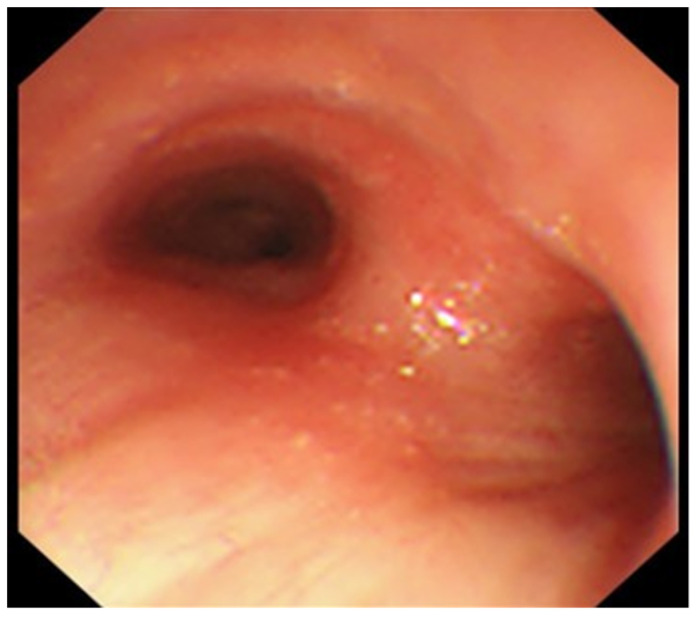
The bronchoscopy image shows a dull formation of the carina.

**Figure 4 medicina-59-00677-f004:**
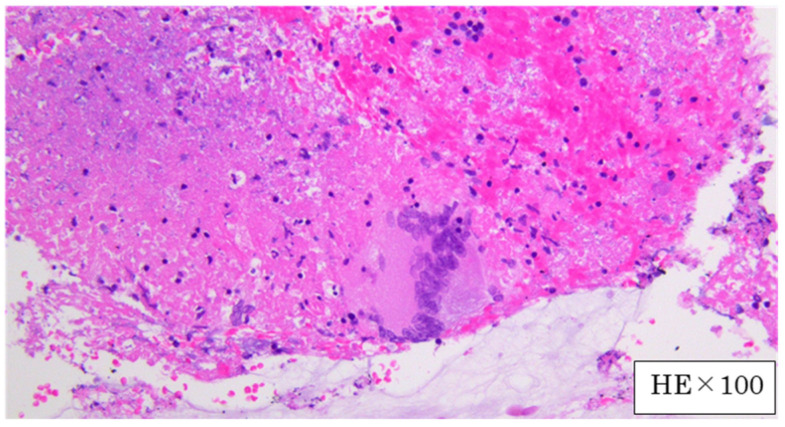
Histological findings of a lymph node obtained with EBUS-TBNA demonstrated epidermoid cell granulomas with Langerhans’ giant cells.

**Table 1 medicina-59-00677-t001:** Laboratory findings (data at the time of transfer to our hospital).

Hemotology	Result	Reference Range	Biochemistry	Result	Reference Range
WBC	23,100/μL	(3300–8600/µL)	AST	10 IU/L	(13–30 IU/L)
Neu.	80.2%	(49.7–72.7%)	ALT	7 IU/L	(7–23 IU/L)
Eos.	2.2%	(0.0–5.0%)	ALP	205 IU/L	(106–322 IU/L)
Bas.	0.3%	(0.0–3.0%)	LDH	171 IU/L	(124–222 IU/L)
Mon.	4.8%	(1.7–8.7%)	BUN	63.4 mg/dL	(8–20 mg/dL)
Lym.	5.3%	(24.5–38.9%)	Cre	9.72 mg/dL	(0.46–0.79 mg/dL)
Hb	9.3 g/dL	(11.6–14.8 g/dL)	TP	7.2 g/dL	(6.6–8.1 g/dL)
PLT	33.3 × 10^4^/µL	(15.8–34.8 × 10^4^/µL)	ALB	4.9 g/dL	(4.1–5.1 g/dL)
			CRP	29.13 mg/dL	(≤0.14 mg/dL)
Serology			Na	134 mEq/L	(138–145 mEq/L)
CYFRA	7.6 ng/mL	(≤2.2 ng/mL)	K	3.5 mEq/L	(3.6–4.8 mEq/L)
NSE	13.3 ng/mL	(≤15.0 ng/mL)	Cl	95.6 mEq/L	(101–108 mEq/L)
PRO-GRP	345 pg/mL	(<81.0 pg/mL)	Coagulation		
sIL-2R	3140 U/L	(190–650 U/L)	D-dimer	2.7 µg/mL	(<1.0 µg/mL)
ACE	17.6 U/L	(7.7–29.4 U/L)	PT	12.6 s	(9.0–13.0 s)
			APTT	42.1 s	(23–40 s)

## Data Availability

Data sharing is not applicable to this article as no datasets were generated or analyzed during the current study.

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
