# Peer review of "A Case of Mediastinal Tuberculous Lymphadenitis in a Chronic Dialysis Patient Diagnosed by Endobronchial Ultrasound-Guided Transbronchial Needle Aspiration (EBUS-TBNA)"

_medicina, 2023, doi:10.3390/medicina59040677_

Round 1

Reviewer 1 Report

A very good presentation of the efficiency of EBUSTBNA in aiding the diagnosis of mediastinal pathology and a reminder of the lurking Mycobacterium in cases withmarkers of chronic infection and lymphathenopathy. The subject of this paper is relevant to the journal's orientation.

Overall it is a great work and I believe is of benefit for the readers of the Journal.

However, I would suggest the supporting bibliographic references to be replaced by more contemporary (references 6, 7 and 8). 

Reviewer 2 Report

The authors present an interesting case of mediastinal tuberculous lymphadenitis. Although it is of interest to the general public, it needs minor improvements before it can be accepted for publication.

1. page 1 under Case: how long did this patient have chronic renal failure? Since the age of 51? 

3. page 3 table: I would advise including the reference value for all measurements between brackets. This makes it possible for the reader to compare them with the values found.

2. page 4 under discussion: the authors state that since 1970 extrapulmonary tuberculosis has been reported more frequently in immunocompromised. Why is this? The author should explain this in a few lines.

3. Page 4 under discussion: regards the statement that there is a high proportion of extrapulmonary tuberculosis in dialysis patients. Why is this? The authors should explain this.

4. page 4 under discussion: the authors write about the advantages of EBUS-TBNA but are there any disadvantages the reader should know?
